# Mismatch between Clinical–Functional and Radiological Outcome in Tibial Plateau Fractures: A Retrospective Study

**DOI:** 10.3390/jcm12175583

**Published:** 2023-08-27

**Authors:** Markus Bormann, David Bitschi, Claas Neidlein, Daniel P. Berthold, Maximilian Jörgens, Robert Pätzold, Julius Watrinet, Wolfgang Böcker, Boris Michael Holzapfel, Julian Fürmetz

**Affiliations:** 1Department of Orthopedics and Trauma Surgery, Musculoskeletal University Center Munich (MUM), University Hospital, LMU Munich, 81377 Munich, Germany; 2Department of Trauma Surgery, Trauma Center Murnau, 82418 Murnau, Germany

**Keywords:** tibial plateau fracture (TPF), Rasmussen score, clinical outcomes, radiological outcomes

## Abstract

Background: The evaluation of tibial plateau fractures (TPF) encompasses the assessment of clinical–functional and radiological parameters. In this study, the authors aimed to investigate the potential correlation between these parameters by utilizing both the clinical–functional and the modified radiological Rasmussen score. Methods: In this retrospective monocentric study conducted at a level-I trauma center, patients who underwent surgery between January 2014 and December 2019 due to a TPF were included. The clinical–functional Rasmussen score prior to the injury, at 1-year postoperatively, and during the last follow-up (minimum 18 months) was assessed using a standardized questionnaire. Additionally, the modified radiological Rasmussen score was determined at the 1-year postoperative mark using conventional radiographs in two planes. Results: A total of 50 patients were included in this study, comprising 40% (n = 20) men, and 60% (n = 30) women, with an average age of 47 ± 11.8 years (range 26–73 years old). Among them, 52% (n = 26) had simple fractures (classified according to Schatzker I–III), while 48% (n = 24; according to Schatzker IV–VI) had complex fractures. The mean follow-up was 3.9 ± 1.6 years (range 1.6–7.5 years). The functional Rasmussen score assessed before the injury and at follow-up showed an “excellent” average result. However, there was a significant difference in the values of complex fractures compared to before the injury. One year postoperatively, both the clinical–functional score and the modified radiological score demonstrated a “good” average result. The “excellent” category was more frequently observed in the functional score, while the “fair” category was more common in the radiological score. There was no agreement between the categories in both scores in 66% of the cases. Conclusions: The data from this retrospective study demonstrated that patients with TPF are able to achieve a nearly equivalent functional level in the medium-term after a prolonged recovery period, comparable to their pre-injury state. However, it is important to note that the correlation between clinical–functional and radiological parameters is limited. Consequently, in order to create prospective outcome scores, it becomes crucial to objectively assess the multifaceted nature of TPF injuries in more detail, both clinically and radiologically.

## 1. Introduction

The incidence of tibial plateau fractures (TPF) has increased significantly over the past decade [1]. Consequently, the treatment strategies for this complex injury have undergone changes. Nowadays, computer tomography (CT) imaging is considered the gold standard for diagnostics [2], leading to the development of novel classification systems [3] and the establishment of a 360° operative treatment [4,5].

The fundamental principles of osteosynthetic treatment aim to achieve the most accurate possible joint surface reduction and anatomical reconstruction of both the width of the tibial head, joint angles, and limb alignment. In 1973, Rasmussen described how these parameters significantly impact patient outcomes [6], a finding that was subsequently validated by Kraus et al. and Beisemann et al. in the past years [5,7]. Additionally, Rasmussen developed a clinical–functional outcome score that is not reliant on radiological parameters [6]. As both scores were shown to be reliable and reproducible, they are still used today to assess the outcome in patients following TPF [8,9,10,11].

In the current literature, short- to medium-term outcomes following osteosynthesis of TPF are described as good to excellent [7,11,12,13]. However, in the long term, the functional scores tend to be lower on average, and the athletic level is reduced compared to pre-injury levels [5,14]. The rate of post-traumatic arthritis (PTA) following TPFs is reported to be between 13 and 83%, which may be higher in patients with articular sided complex fractures. Consequently, approximately 7% of the patients require a secondary total knee arthroplasty (TKA) within 10 years post-fracture [14,15,16]. However, there remains a scarcity in the literature reporting on functional outcomes and their correlation to fracture morphology.

The aim of this study is (1) to report on functional outcomes in patients following TPF and (2) to correlate them with the radiological outcomes. Hypothesis (1) was that after TPF, patients would achieve functional values equivalent to their pre-injury functional values and hypothesis (2) was that there is a correlation between functional outcomes, fracture morphology, and anatomical reconstruction.

## 2. Materials and Methods

### 2.1. Patient Selection

A retrospective chart review was performed on all patients at a German level-I trauma center, who underwent surgery for TPF between January 2014 and December 2019. Institutional review board approval was obtained before the initiation of the study. Patients were included if they had confirmed intra-articular TPF during pre-operative CT scans, if they were aged > 18 years, and if they had detailed documentation about trauma mechanism and information on demographics such as gender and age. Furthermore, radiographic imaging (X-ray in anteroposterior and lateral view) 12 months after surgery was required. Minimum follow-up was set at 18 months. Patients were excluded if they had extraarticular fractures (AO/OTA 41-A), other fractures than TPF, tibial shaft fractures, as well as inconsistent documentation.

### 2.2. Surgical Technique

Patients with TPF were operated on either by open reduction and internal fixation (ORIF) or by arthroscopically assisted closed reduction and internal fixation (CRIF).

### 2.3. Postoperative Rehabilitation

All patients were treated with a standardized, clinic-specific postoperative protocol. This includes an 8-week partial load-bearing period as well as a hard frame orthesis with flexion limitation at 60 degrees for 6 weeks.

### 2.4. Clinical Analysis

Outcome analysis included the clinical–functional Rasmussen score (Table 1). This score was collected for the period directly before sustaining TPF, 1 year postoperatively, as well as for the minimum follow-up. Additionally, at final follow-up, all patients were assessed for passive and active range of motion and clinical laxity testing.

### 2.5. Radiographic Analysis

The fractures were classified using the established systems of Schatzker, AO/OTA, and Moore by the first and senior author (Consultant and head of department, respectively) as well as by 2 scientific assistants on CT scans. Discrepancies in classifications between the raters were solved by discussion. The modified radiological Rasmussen score was determined at 1 year postoperatively using conventional X-rays in two planes by the same research group (Table 1). Fractures were classified as simple fractures when they had a confirmed TPF according to Schatzker I-III. In contrast, fractures were classified as complex fractures when they had a confirmed TPF according to Schatzker IV-VI and/or radiological evidence of knee dislocation according to Moore [17,18].

### 2.6. Statistical Analysis

Descriptive statistics were summarized as means and standard deviations for quantitative variables and counts and frequencies for categorical variables. The significance of differences in means and frequencies of continuous and categorical variables was examined. For this purpose, the Mann–Whitney, Wilcoxon, and McNemar tests, and the Spearmen correlation coefficient were used. Statistical significance for all comparisons was set at *p* < 0.05. All analyses were performed with SPSS Statistics 26.0 (IBM Corp., Armonk, NY 10504, USA). The graphical representation was performed using SPSS Statistics 26.0 (IBM Corp., Armonk, NY 10504, USA) and Microsoft Excel 365 MSO Version 2207 (Microsoft Corp., Redmond, WA, USA).

### 2.7. Rasmussen Scores

**Table 1 jcm-12-05583-t001:** Rasmussen scores—criteria and evaluation.

Radiological Score		Pts	Clinical–Functional Score		Pts
**Depression**	None	6	**Pain**	No pain	6
	<5 mm	4		Occasional pain	5
	5–10 mm	2		Stabbing pain in certain positions	4
	>10 mm	0		Constant pain after activity	2
**Condylar widening**	None	6		Significant rest pain	0
	<5 mm	4	**Walking capacity**	Normal for age	6
	5–10 mm	2		Outdoor > 1 h	4
	>10 mm	0		Outdoor > 15 min	2
**Angulation (varus/valgus)**	None	6		Only indoors	1
	<10°	4		Immobile	0
	10–20°	2	**Extension**	Normal	6
	>20°	0		Lack of extension < 10°	4
				Lack of extension > 10°	2
			**Range of motion**	>140°	6
				>120°	5
				>90°	4
				>60°	2
				>30°	1
				>0°	0
			**Stability**	Normal stability	6
				Instability in 20° flexion	5
				Instability in extension <10°	4
				Instability in extension >10°	2
**Radiological Score**		**Clinical–Functional Score**		**Evaluation**	
18 points		27–30 points		excellent	
12–17		20–26		good	
6–11		10–19		fair	
0–5		4–9		poor	

## 3. Results

### 3.1. Participants

In this monocentric study, 319 patients were treated for TPF between January 2014 and December 2019. Of these patients, 50 were eligible for inclusion in the study (Figure 1).

The mean age of the patients was 47 ± 11.8 years, with a range between 26 and 73 years old. The mean follow-up was 3.9 ± 1.6 years, with a range between 1.6 and 7.5 years. Overall, 26 patients could be assigned to simple fractures (according to Schatzker I–III), while 24 patients were diagnosed with complex fractures (according to Schatzker IV–VI). The patient-specific data are presented in Table 2.

### 3.2. Surgical Technique

A total of 94% (n = 47) of patients were treated by ORIF, while 6% (n = 3) received arthroscopic-assisted CRIF with screw osteosynthesis. Of these 47 ORIF patients, 76.6% (n = 36) were treated by a single approach, most frequently anterolateral (66%, n = 31), while 23.4% (n = 11) received combined approaches. Single plate osteosynthesis was performed in 70.2% (n = 33, most common anterolateral—84.8%, n = 28) and 29.8% (n = 14) received combined osteosynthesis (double/triple plate, plate + screws). A total of 17% (n = 8) of patients treated by ORIF also received additional knee fracturoscopy.

Furthermore, in 36% (n = 18) of patients concomitant meniscal and/or ligamentous injury were treated in addition to osteosynthesis. The injuries treated were anterior/posterior cruciate ligament (ACL/PCL) refixations, meniscus sutures and collateral ligament refixations.

### 3.3. Clinical Outcomes

Table 3 shows the values of the Rasmussen scores at the different survey time points, compares the simple and complex fractures according to Schatzker, and lists the most frequent variant for each assessment category (pain, walking capacity, extension, etc.).

In the clinical–functional Rasmussen score, patients achieve an average score before injury, which corresponds to an “excellent” result according to Rasmussen. One-year post-surgery the mean score corresponds to a “good” value for both simple and complex fractures. However, it is significantly worse in both groups (*p* < 0.001) compared to the pre-injury scores. Although the difference between the groups is measurable at this point, it is not statistically significant (*p* = 0.052). As the follow-up progresses, both groups demonstrate an increase in the average score, eventually reaching an “excellent” score. However, the value achieved for complex fractures remains significantly worse (*p* < 0.01) than before the injury. Otherwise, there is no significant difference (*p* = 0.071) for the simple fractures at the final follow-up.

The modified radiological Rasmussen score, one year after surgery, indicates a “good” result for both simple and complex fractures. The difference between the groups is not significant at this point (*p* = 0.447).

Figure 2 shows the number of patients in each result group. One year after surgery, more patients in the clinical–functional score group showed an “excellent” result compared to the modified radiological score group (*p* = 0.189). Notably, there are significantly more patients rated as “poor/fair” radiologically (n = 12) than clinical–functional (n = 3) at this time (*p* = 0.035).

Figure 3 shows that the position of the median for both scores is within the “good” outcomes group 1-year postoperatively. In each case, the median is located above the arithmetic mean. Additionally, there is a noticeable reduction in scatter for the clinical–functional score leading up to the follow-up.

Table 4 shows the clinical–functional and radiological Rasmussen score after one year in a cross-tabulation.

When analyzing the assignment of patients to their respective outcome groups (poor, fair, good, excellent) based on the clinical–functional and modified radiological score after one year, it was found that 66% (n = 33) of the 50 cases had no match. Thereby, 50% (n = 25) of the patients had a lower rating in the radiological score compared to the clinical–functional score, while 16% (n = 8) showed a higher rating in the radiological score. The Spearman correlation coefficient shows no relevant correlation for the two scores (Rho = 0.075).

In the subgroup of patients who scored “moderate,” there was entirely no agreement (in 100% of the cases) with the other score. Regarding patients rated as “good” in the clinical–functional score, 53.3% (n = 16) had no radiological match, and within this group 68.8% (n = 11) displayed a worse radiological score. On the other hand, among patients rated as “good” radiologically, 50% (n = 14) did not exhibit a corresponding result in the clinical–functional score, and within this group 92.9% (n = 13) had a better clinical–functional rating.

Interestingly, three patients with a clinical–functional rating of “fair” achieved an “excellent” radiological score twice and a “good” score once simultaneously. In the patient with a “good” rating, only a depression in the articular surface of < 5 mm was observed radiologically. These three patients shared the characteristic of exhibiting instability, in addition to individual differences in the clinical–functional score.

Among the twelve patients rated as “poor” or “fair” (Figure 2) in the radiological score, they either showed a significant depression exceeding 10 mm and/or a condylar widening ranging from 6 to 10 mm. In contrast, eleven patients achieved a “good” rating, while one patient achieved an “excellent” rating in the clinical–functional score.

Figure 4 shows the X-ray in two planes of a 31-year-old female patient 1 year postoperatively with a poor radiological Rasmussen and a good functional Rasmussen score.

## 4. Discussion

The most important finding of this study was that patients with TPF demonstrated an “excellent” outcome at a mean of 3.9 (+1.6) years post-surgery, as measured by the clinical–functional Rasmussen score. This outcome was observed regardless of the severity of the bony injury, according to the Schatzker classification. However, it is noteworthy that the clinical–functional scores were significantly worse after one year, but gradually improved during the subsequent observation period, indicating a prolonged recovery. 

One year postoperatively, the patients achieve an average “good” score on both the clinical–functional and modified radiological Rasmussen score. However, this work also demonstrated that the different outcome groups (poor, fair, good, excellent) do not match in most of the cases, especially in the worse results. This underlines the importance of accurately assessing clinical function independently of postoperative radiographic findings for further treatment recommendations. Additionally, this once again proves that TPF is a complex joint injury that extends beyond just a fracture.

Previous research has reported a conversion rate of 3–7% for TKA within the first five years following osteosynthetic treatment of TPF [15,19,20], with the highest risk occurring within the initial two years [21,22,23]. Therefore, when discussing the possibility of secondary TKA with patients, it is crucial to consider the extended recovery period and the individual knee function independently of the X-ray. Moreover, it is important to note that TKA outcomes for patients with post-traumatic arthritis (PTA) are inferior, and the complication rates are higher compared to primary gonarthrosis cases [24,25].

In 1973, Rasmussen introduced his clinical–functional score [6]. The subjectively assessed parameters such as pain, walking capacity, and instability outweigh the objectively recorded ones like extension and range of motion, which is notably a limitation of the clinical–functional Rasmussen score. In particular, instability, which has been identified as a significant factor in the development of post-traumatic osteoarthritis [16], can also be evaluated through a clinical–apparative examination [26]. The fact that this study’s patients showed an increase in BMI and a decrease in activity level during the recovery period suggests that the subjectively perceived excellent outcome may not be objectively substantiated. As demonstrated in this case for complex fractures, a statistically significant decrease in score (when comparing pre-injury to post-injury) does not necessarily result in a change in the scoring category. Hence, it is crucial to question this categorization.

Rasmussen’s radiological score was also first described in 1973 [6]. Since then, significant advancements have occurred in radiological diagnostics for TPF, pre-, intra-, and postoperatively. Preoperative CT imaging is now considered the gold standard, and postoperative CT imaging is widely used for reposition control [2,27]. CT imaging provides more accurate visualization of the parameters used in the Rasmussen score, including depression, angulation, and widening of the tibial plateau [28,29,30]. This has led to well-defined limits for angulation and widening of the tibial plateau [2,6,31,32,33]. Different threshold values exist for the joint step, depending on whether it is in the load-bearing and/or meniscus-covered part. However, the current threshold values discussed are significantly lower than the gradations defined by Rasmussen [6,7,31,34,35,36,37,38].

In recent years, several clinical/functional outcome scores have been established, such as KOOS, Tegner, and IKDC, some of which are more comprehensive than the score developed by Rasmussen [39,40,41,42]. These scores mostly rely on subjective parameters [39,40,41,42]. However, apart from the modified Rasmussen score, no other radiological score has been widely adopted. Consequently, both Rasmussen scores are still frequently used in the current literature [8,9,10].

The lack of clear recommendations for MRI imaging in TPF indicates that the focus of radiological imaging continues to be the assessment of bony injury [2].

With the improved understanding of TPF as a complex joint injury in recent years, it has become more evident that, in addition to the bony and functional parameters defined by Rasmussen, meniscus, cartilage, and soft tissue lesions, and measurable instabilities contribute to the development of PTA and the overall outcome after TPF [31,43,44,45]. Extended imaging techniques (CT and MRI) and instrument-based diagnostics, including dynamic assessment, can help objectify these parameters. It is necessary to develop a scoring system based on comprehensive data that accurately represent the current and future outcomes after TPF.

## 5. Limitations

This study has several limitations. First, the data retrieved from this study are of retrospective nature, which could create selection bias. Second, the follow-up was only 18 months, as no long-term data were available. Third, no control group was available. However, all patients included in this study were indicated for surgery. Fourth, as mentioned above, apart from the modified Rasmussen score, no other radiological score has been widely adopted to date. Consequently, both Rasmussen scores are still frequently used in the current literature. Fifth, knee joint laxity was not measured in this study using dynamic reproducible methods. Lastly, no postoperative MRI was available to assess for progression of osteoarthritis or cartilage defects. 

## 6. Conclusions

The data from this retrospective study demonstrated that patients with TPF are able to achieve a nearly equivalent functional level in the medium-term after a prolonged recovery period, comparable to their pre-injury state. However, it is important to note that the correlation between clinical–functional and radiological parameters is limited. Consequently, in order to create prospective outcome scores, it becomes crucial to objectively assess the multifaceted nature of TPF injuries in more detail, both clinically and radiologically.

## Figures and Tables

**Figure 1 jcm-12-05583-f001:**
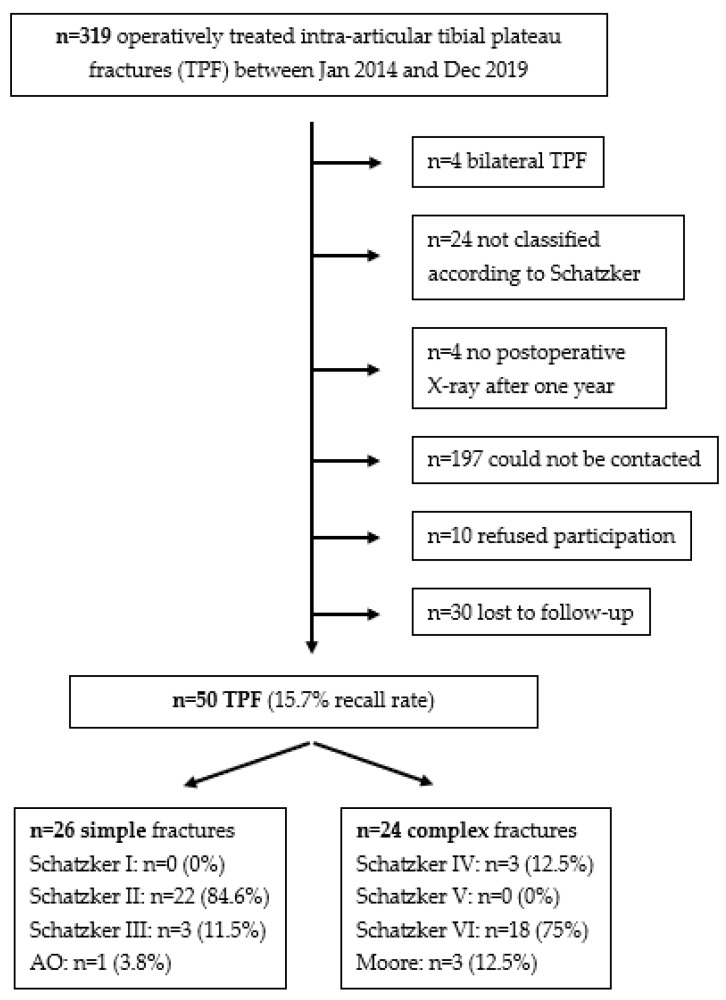
Flow chart patient selection.

**Figure 2 jcm-12-05583-f002:**
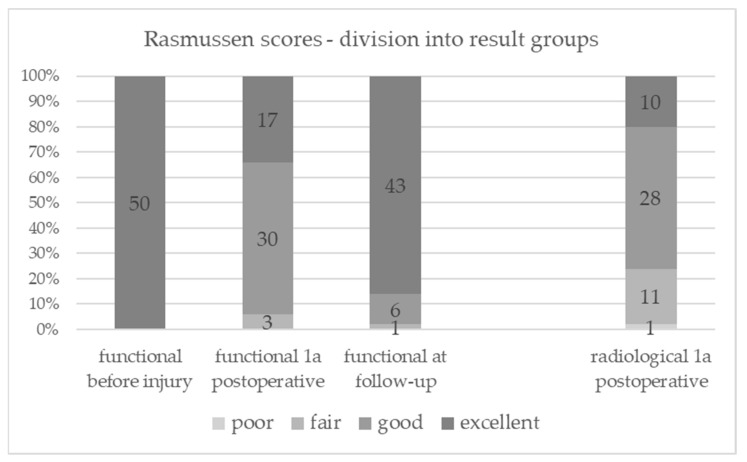
Rasmussen scores—division into result groups.

**Figure 3 jcm-12-05583-f003:**
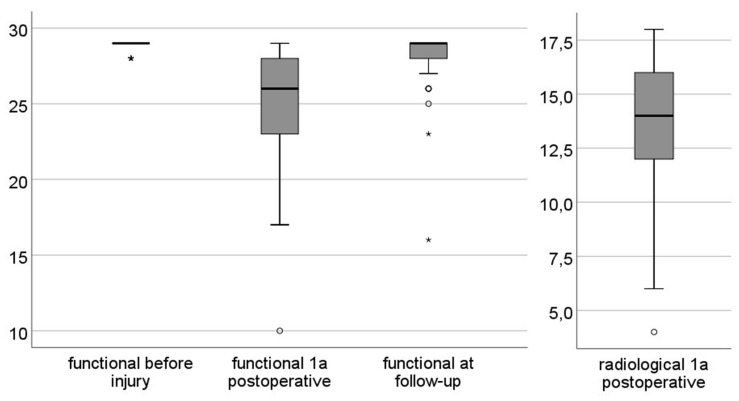
Dispersion of Rasmussen scores with median position. * = values with an interquartile range more than 3.

**Figure 4 jcm-12-05583-f004:**
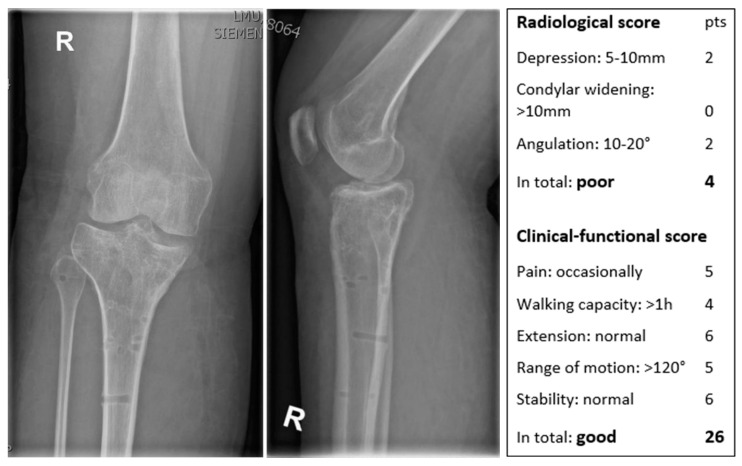
X-ray in two planes of a right (R) knee with the scores of a 31-year-old female patient.

**Table 2 jcm-12-05583-t002:** Patient-specific data total collective.

Criteria	Total Collective (n = 50)	*p*-Value
Men vs. women	40% (n = 20) vs. 60% (n = 30)	
Mean age	47 ± 11.8 years (range 26-73 years old)	
Mean follow-up	3.9 ± 1.6 years (range 1.6-7.5 years)	
Mean BMI at surgery BMI at final follow-up Difference	24.4 ± 3.5 25.2 ± 3.6 +0.8	0.001
Schatzker (n) I II III IV V VI AO/Moore	0 (0%) 22 (44%) 3 (6%) 3 (6%) 0 (0%) 18 (36%) 4 (8%)	
Surgical technique - Knee arthroscopy - ORIF	3 (6%) 47 (94%)	

**Table 3 jcm-12-05583-t003:** Rasmussen scores—simple vs. complex—most common assessment category.

Criteria	Total Collective (n = 50)	*p*-Value
Rasmussen functional before injury	28.84 ± 0.37 (excellent)	
simple vs. complex	28.77 vs. 28.92	0.16
-Pain	84% (n = 42) no pain	
-Walking capacity	100% (n = 50) normal	
-Extension	100% (n = 50) normal	
-Range of motion	100% (n = 50) >120°	
-Stability	100% (n = 50) normal	
Rasmussen functional 1a postoperative	24.68 ± 3.61 (good)	
simple vs. complex	25.69 vs. 23.58	0.052
-Pain	76% (n = 38) occasional	
-Walking capacity	44% (n = 22) normal	
-Extension	54% (n = 27) normal	
-Range of motion	66% (n = 33) >120°	
-Stability	88% (n = 44) normal	
Rasmussen functional at follow-up	28.0 ± 2.17 (excellent)	
simple vs. complex	28.35 vs. 27.63	0.489
-Pain	80% (n = 40) no pain	
-Walking capacity	88% (n = 44) normal	
-Extension	88% (n = 44) normal	
-Range of motion	90% (n = 45) >120°	
-Stability	94% (n = 47) normal	
Rasmussen radiological 1a postoperative	13.44 ± 3.64 (good)	
simple vs. complex	14.0 vs. 12.83	0.447
-Depression	38% (n = 19) None	
-Condylar widening	46% (n = 23) None	
-Angulation	54% (n = 27) None	
Rasmussen functional vs. radiological 1a postoperative	Spearman-Rho = 0.075	0.605

**Table 4 jcm-12-05583-t004:** Cross-tabulation Rasmussen scores 1-year postoperatively.

		Rasmussen Radiological 1a Postoperative				In Total
		Excellent	Good	Fair	Poor	
Rasmussen functional 1a postoperative	excellent	3	13	1	0	17
	good	5	14	10	1	30
	fair	2	1	0	0	3
In total		10	28	11	1	50

## Data Availability

Not applicable.

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
