# Peer review of "Mismatch between Clinical–Functional and Radiological Outcome in Tibial Plateau Fractures: A Retrospective Study"

_jcm, 2023, doi:10.3390/jcm12175583_

Round 1

Reviewer 1 Report

This is a debated topic and could be interesting for readers.

The study is well-written, and the sections provide quite good background.

Moreover, the paper has some limitations:

1-    no control group

2-    the follow-up is only 18 months

3-    Only the modified Rasmussen score was used to radiological evaluate each patient

Concluding, this could be -in my opinion- a pilot study that may pave the way for further prospective studies.

For these reasons, I suggest accepting with minor revisions

Comments:

TITLE:

ok

INTRODUCTION:

The introduction is short, please add some lines.

how many aims? 1 or 2? If 2 aims are declared, you need to also explain 2 hypotheses. Please adjust this section.

Level of evidence?

METHODS:

L76 Did X-rays were taken only at 12 months of follow-up?

L82, how many surgeons did perform the surgery? are all expert surgeons?

L91 how many months was set for the final follow-up?

DISCUSSION:

L229 post-traumatic

Reviewer 2 Report

line 146-154. I get the sense that this paragraph is the "crux" of the paper, but I'm not exactly sure what it is trying to say.  How were pre-injury scores obtained? Is the point that "excellent" is the goal? How long does it take for both groups to normalize to excellent? 

what time point is 1a (1 year anniversary?) should be clarified 

overall quite good, I think some of the syntax just needs altering in the paragraph noted above
